# Sex-Specific Multimorbidity–Multibehaviour Patterns in Primary Care Populations

**DOI:** 10.3390/ijerph22040485

**Published:** 2025-03-24

**Authors:** Konstantinos Spyropoulos, Naomi J. Ellis, Christopher J. Gidlow

**Affiliations:** 1Centre for Health and Development (CHAD), University of Staffordshire, Stoke-on-Trent ST4 2DF, UK; n.j.ellis@staffs.ac.uk; 2School of Medicine, Keele University, University Road, Staffordshire ST5 5BG, UK; c.gidlow@keele.ac.uk; 3Research and Innovation Department, Midlands Partnership University NHS Foundation Trust (MPFT), St Georges Hospital, Corporation Street, Stafford ST16 3AG, UK

**Keywords:** multimorbidity, multibehaviours, exploratory factor analysis, primary care, sex-specific multimorbidity patterns

## Abstract

Background: A conceptual shift in healthcare emphasises multimorbidity and multibehaviours as interconnected phenomena, highlighting dose–response associations and sex-specific differences. Data-driven approaches have been suggested for overcoming methodological challenges, of multimorbidity research. By using exploratory factor analysis, this study aimed to identify sex specific lifestyle associative multimorbidity patterns, providing valuable evidence to primary care providers and informing future multimorbidity guidelines. Methods: A retrospective observational study examined the electronic health records of three general practices in the UK between 2015 and 2018. The participants were aged 18+ with lifestyle multimorbidity, having engaged with multiple health risk behaviours. Stratified exploratory factor analysis with oblique rotation was used to identify sex specific lifestyle associative multimorbidity patterns. Results: The study included N = 7560 patients, with females comprising 53.9%. Eight independent lifestyle associative multimorbidity patterns were identified and distributed as follows. For females, three patterns emerged: cardiometabolic–neurovascular spectrum disorders (42.97% variance), respiratory conditions (8.08%), and sensory impairment (5.63%), with 25.4% assigned to these patterns. For males, five patterns were revealed: cardiometabolic–vascular spectrum disorders (34.10%), genitourinary (9.19%), respiratory–vision (8.20%), ocular (5.70%), and neurovascular–gastro–renal syndrome (4.54%), with 43%. Conclusions: We revealed eight different sex-specific lifestyle-associated patterns, implying the need for tailored clinical approaches. The application of exploratory factor analysis yielded clinically valuable and scientifically rigorous multimorbidity patterns. Clinically, the findings advocate for a paradigm shift towards person-centred care, integrating multimorbidity and SNAP multibehaviours to enhance the complexity of inquiry and treatment of high-risk populations.

## 1. Introduction

A recent conceptual shift in the theoretical landscape of healthcare acknowledges that health phenomena, including morbidities and health risk behaviours (HRBs) such as smoking, nutrition, alcohol intake, and physical activity (SNAP), often co-occur in identifiable clusters beyond mere chance rather than existing in isolation. This evolution has led to the emergence of concepts such as multimorbidity (MM), referring to the presence of multiple concurrent chronic conditions [1], and multibehaviours (MB), referring to engagement in multiple health risk behaviours [2].

Both health phenomena have garnered attention in curative and preventive medicine. Prominent figures in the scientific community, such as Prochaska [2], have long since advocated for a shift in the current healthcare system to incorporate ideas from both behavioural and clinical paradigms. Loprinzi [1] proposed a unified multimorbidity–multibehaviour (MM-MB) theoretical and clinical framework.

Lifestyle MM emerged, with studies providing support for the synergistic health effect of SNAP-MB revealing a strong dose–response association between the number of involved SNAP-HRBs and MM risk [3,4,5,6], but also sex-specific differences in the relationship between MM and SNAP-HRBs. Specifically, men and women were found to exhibit different thresholds for developing MM based on the number of SNAP-HRBs or the definition of MM used [7].

However, methodological challenges (due to the complexities of MM) obscure a deeper scientific and clinical understanding and introduce subjectivity to researcher and clinician decisions [7], for example, determining the number of morbidities to include [8], setting the minimum number of chronic conditions [9,10], defining cut-offs (e.g., MM2+ or MM3+), or employing accumulative indices [11].

Accordingly, healthcare provision for people with MM remains anchored to a one-size-fits-all, single-disease approach. Clinicians continue to rely on guidelines designed to address single diseases, which jeopardises the delivery of optimal care [12,13,14]. Such an approach fails to address critical questions imposed via MM research, such as the potential sex-specific MM-MB patterns, and the need for tailored treatment responses. A consequence of this systemic failure is the overexposure of patients with MM to pharmaceutical treatments and risks of adverse medication effects associated with polypharmacy [15].

Responding to these challenges, researchers moved away from using simplistic counts and weighted indices towards advanced data-driven methods to achieve a deeper clinical understanding [8,12,16,17]. This shift redirects analytical inquiry towards concepts like “associative MM” [18], “causal MM” [19], or “cluster medicine” [20], all of which relate to the identification of non-random linkages between morbidities.

By merging the constructs of lifestyle and associative MM under a unified examination, the present study explored the distinct sex-based clustering of morbidities [21] with shared underlying pathophysiological pathways—such as chronic inflammation, oxidative stress, and metabolic syndrome—linking morbidity clusters (MM) both to each other and to common risk factors (MB) [22]. Based on the premise that morbidities can systematically correlate beyond randomness [23], it is expected that data-driven approaches, such as cluster analysis (CA) and exploratory factor analysis (EFA), can provide insight into the synergetic effects of MB on specific MM patterns [18].

This integrative perspective offers promise for overcoming the limitations of prevailing definitions of MM, such as the commonly used criterion of acquiring two or more chronic conditions in the same individual (MM2+) [23]. By providing a holistic framework, we can enhance the study of MM. Following this causal analytical pathway, the present study aims to contribute clinically useful knowledge to support the development of tailored sex-specific MM guidelines [21]. We go beyond simply cataloguing ad hoc concomitant morbidities, such as those seen in concurrent MM analyses, or reporting statistically significant associations without causal justification, as observed in the simple cluster analysis of morbidities [22]. Instead, the study seeks to deepen understanding of the synergistic effects between chronic conditions and, ultimately, to guide the development of future MM guidelines [21]. The present study aims to identify sex-specific lifestyle- associative multimorbidity patterns exploring how clusters of morbidities emerge differently in men and women when they engage in multiple SNAP-HRBs (smoking, bad nutrition, excessive alcohol consumption, and physical inactivity).

## 2. Materials and Methods

### 2.1. Study Design and Setting

This is a retrospective observational study based on data (from 2015–2018) retrieved from electronic health records (EHRs) from three general practices (GPs) in the UK. Ethical approval was obtained from the NHS Health Research Authority (East of England—Essex Research Ethics Committee). The STROBE checklist for observational studies was used to guide reporting (Appendix A).

### 2.2. Participants

The sample comprised all patients registered at the participating general practices who were aged 18+ years, had developed an MM (MM2+) from the list of 40 morbidities [24], and were found to engage with at least two SNAP-MB (n = 7560). This study used a subset of a larger dataset of 21,079 registered patients, which was originally used to examine the broader association between SNAP-MB and multimorbidity (MM) under various operational definitions. The analysis was stratified by sex and adjusted for multiple sociodemographic variables.

Missing data in the original dataset was addressed using multiple imputation (MI) in SPSS. Briefly, the proportion of missing data varied, with some variables related to the included morbidities having no missing values, while the nutrition variable showed a high missing rate of 55.6%. The missing values for other SNAP-HRBs were also notable: smoking at 7.6%, alcohol consumption at 26.1%, and exercise at 39.3%. In terms of demographic information, the missing data ranged from nearly zero for deprivation to 23.4% for ethnicity, with the employment variable exhibiting an extremely high missing rate of 85.6% within a sample of 21,079 participants.

To ensure optimal results for the imputed values, all of the auxiliary variables were incorporated into the multiple imputation (MI) process. The Markov Chain Monte Carlo (MCMC) method using a logistic regression model was employed, as there was no monotonicity and the variables were categorical. This process involved ten iterations, where SPSS produced five imputed datasets. By applying “Rubin’s rules”, a pooled dataset was obtained. Applying a logistic regression analysis on the four SNAP-HRBs, a reasonable comparison between the imputed and observed values was applied. All statistical analyses were carried out on the pooled imputed dataset.

### 2.3. Data Processes and Variables

Anonymised data were extracted from general practices by the NHS Commissioning Support Unit, which also provided support with the appropriate translation of Read codes during the extraction process to identify morbidities of interest. This approach addressed concerns experienced by similar studies [18,23] regarding whether or not GP personnel had the experience to correctly use the patients’ EHRs. The variables used in the present study are summarised below.

The sociodemographic variables included age and sex at birth (male, female). Moreover, the present study utilised the most commonly operational definition of multimorbidity, which regards the acquisition of two or more chronic conditions from the same individual (MM2+). Due to the lack of a standardised method for measuring multimorbidity, the study adopted the methodology used in a well-known study by Barnett et al. [24], which includes a list of 40 different morbidities encompassing a wide range of both physical and mental health conditions. This list fulfils the minimum inclusion criteria established by two systematic reviews [9,10], which serve as essential benchmarks for any multimorbidity assessment. These reviews indicate that a comprehensive multimorbidity study should incorporate a minimum of 11 or 12 of the most prevalent chronic conditions, such as cancer, diabetes, depression, hypertension, myocardial infarction, chronic ischaemic heart disease, arrhythmias, heart failure, stroke, COPD, and arthritis.

Finally, multibehaviours were examined via the information extracted regarding the four most common SNAP-HRBs: smoking, nutrition, alcohol intake, and physical activity. Considering the limitations in recording these data within primary care and to gain an accurate picture of how patients engage with SNAP health behaviours, we also examined the EHRs to gather information regarding whether these behaviours were present via any form of documentation, including cases where patients received guidance on modifying these behaviours.

To support both the theoretical and pragmatic approach of the present study, the binary categorisation of the present SNAP-HRBs was applied since it is hypothesised to better capture the cumulative exposure to the specific risk behaviours, facilitating the examination of their association with the applied multimorbidity index. The binary variables were smoking status—“ever-smoker” and “never-smoker” based on current status; nutrition—“poor diet” (poor or average) and “good diet”; alcohol—“normal consumption” (within recommended daily limits or never drinking) and “excessive” (above recommended daily limits); physical activity—“physically inactive” (moderately active or inactive) and “active” (meeting the recommended guidelines).

Where data on the behaviours under investigation were missing, evidence of advice or intervention by healthcare providers that related to those behaviours was used (e.g., “patient advice about exercise” would be used to define them as “physically inactive”). The accumulation of SNAP-MB was calculated as the sum of their dichotomised version (1 for engaging with a single SNAP-HRB and 0 for not engaging) that produced an overall score ranging from 0 to 4. 

### 2.4. Statistical Analysis

Exploratory factor analysis was applied to analyse the correlations between morbidities to reveal possible associative MM patterns of those suffering from MM2+ who engaged with SNAP-MB. Principal axis factoring with oblique rotation was chosen as a method for two reasons. First, it has been acknowledged that the extracted patterns are limited, thus being unable to fully explain the total variance when examining morbidities. Second, oblique rotation allows the extracted factors to be associated with each other, which is appropriate here where morbidities could be associated [18], even if where a specific morbidity could be part of more than one MM pattern [22].

Allowing underlying factors to correlate with each other makes interpretation more complex. The only remedy is a thorough examination of the factor loadings of both emerging matrices, namely, the pattern and structure. This is because, when factors are examined independently, factor loadings can simultaneously represent each factor’s correlation and regression coefficients. This indicates the strength of the relationship between the variable and the factor, as well as how much of the variance the specific variable explains within that factor. This property is divided within the pattern and structure matrices [25].

Permitting an underlying association between factors simply means that factor loadings in the pattern matrix provide information about the overall strength of the relationship between each variable on each factor (acting as a regression coefficient), while the information provided within the structure matrix focuses on the unique relationship between each variable and each factor after controlling for other factors (acting as a partial correlation) [25]. Thus, while the pattern matrix’s interpretation is still a given, thoughtful consideration of the structure matrix is suggested, and the display of both matrices increases the transparency of interpretation.

To support further meaningful interpretations, only morbidities with factor loadings over 0.3 were included and interpreted as part of the emergent MM pattern [26]. Factors were extracted only when their eigenvalues exceeded the threshold of 1.0. The extracted factors represent the given MM patterns, and their included morbidity factor loadings represent their contributors [25].

Due to the categorical nature of morbidity variables (0 for no morbidity and 1 for morbidity), a tetrachoric correlation was applied. This is an accepted statistically heuristic approach assuming that, despite being categorical (and, as such, violating assumptions of linearity and normal distribution), the variables under investigation share an underlying continuum with normally distributed properties, e.g., an underlying latent causal morbidity progression that is not directly observable [27].

The sampling adequacy for analysis is verified by the Kaiser–Mayer–Olkin (KMO) measure. According to Field [25], a minimum acceptable basis regarding the goodness of fit is when the KMO value reaches or exceeds the threshold of 0.5. Progressively, values between 0.5 and 0.7 are considered moderate, while values between 0.7 and 0.8 are good, and values of 0.8 and 0.9 or above reflect great and superb goodness of fit, respectively. Finally, both a Kaiser’s criterion >1 and the scree plot inflexion point were considered before judgement was made about the number of factors retained for the final analysis. Given that the sample size of the current study for each investigation significantly surpassed the threshold of 250, any average communality exceeding 0.6 establishes Kaiser’s criterion as a robust measure on its own [25]. To investigate the prevalence of the emergent MM patterns, the MM2+ operational definition was applied; i.e., to allocate a person to a specific MM pattern, a minimum of two of the factor’s included morbidities was necessary. Furthermore, the analysis was conducted separately for females and males. This is because the evidence suggests [23] that sexes might be affected by different MM patterns. This suggests either the existence of different determinants or differences in the magnitude of associations [28]. Finally, to achieve clinically valuable outcomes, only morbidities with a prevalence greater than 1% per sex were included in the study. Three certified doctors (one primary care physician and two hospital specialists) reviewed and verified the clinical value of the emergent MM patterns.

SPSS (version 28) was used for the exploratory factor analysis and data preparation was performed using the open-source software Jamovi 1.6.23 (as SPSS was unable to perform a tetrachoric correlation).

## 3. Results

### 3.1. Sample Characteristics

The sample comprised 7560 patients who had at least two morbidities (MM2+) and engaged with SNAP-MB. Sex was relatively balanced, with females comprising the majority group at 53.9% (n = 4079) and males at 46.1% (n = 3482). The older age group (67+ years) was the largest (40.1%, n = 3032), followed by the middle-aged group (46–66 years, 35.8%, n = 2707) and the younger group (18–45 years, 24.1%, n = 1820).

Table 1 shows the distribution of single morbidities for both sexes. Anorexia–bulimia and multiple sclerosis were excluded from analysis for both sexes since their prevalence was <1%. Parkinson’s disease was also excluded for the same reason for females.

### 3.2. Multimorbidity Patterns

#### 3.2.1. Females

A principal axis factoring (PAF) with oblique rotation (Oblimin) was conducted on 34 morbidities. The sampling adequacy for analysis was verified by the Kaiser–Mayer–Olkin measure. The KMO of 0.808 was of great magnitude, according to Field [25]. Bartlett’s test of sphericity x2(105) = 305.77, *p* < 0.001, indicated that the correlations between items were sufficiently large for PAF. An analysis was run to obtain the eigenvalues for generating factors from the data. Three factors had eigenvalues over Kaiser’s criterion of 1 and collectively explained 56.69% of the variance. The scree plot (Figure 1) showed an inflexion that justified the retaining of three factors contradicting the Kaiser’s criterion. The Kaiser’s average communality was found to be 0.62, which is larger than the threshold of 0.6 that has been set for samples sizes above 250 people [25]. Therefore, all three factors were retained. Table 2 and Table 3 show the factor loadings for both the pattern and structure matrices after the rotation.

The items were clustered under three factors for females. Factor 1 (42.97%), under the unified label of cardiometabolic and neurovascular spectrum disorders, was determined by the associations between coronary heart disease, atrial fibrillation, hypertension, peripheral vascular disease, chronic kidney disease, stroke and transient ischaemic attack, diverticular disease, diabetes, dementia, and cancer. Factor 2 (8.08%), labelled as respiratory conditions, represented COPD and bronchiectasis. Factor 3 (5.63%) was labelled sensory impairment and comprised blindness, glaucoma, hearing loss, and dementia.

Almost a quarter of the sample (25.4%) belonged to at least one of these patterns, with prevalence ranging from 21.4% for the cardiometabolic–neurovascular pattern to 3.2% for sensory impairment and 0.4% for respiratory conditions.

#### 3.2.2. Males

A PAF with oblique rotation (Oblimin) was conducted on 35 morbidities. The sampling adequacy for analysis was verified by the Kaiser–Mayer–Olkin measure. The KMO of 0.680 was found to be of good magnitude, according to Field [25]. Bartlett’s test of sphericity x2 (105) = 280.503, *p* < 0.001, indicated that the correlations between items were sufficiently large for PAF. An analysis was run to obtain the eigenvalues for generating factors from the data. Five factors had eigenvalues over Kaiser’s criterion of 1 and, in combination, explained 61.75% of the variance, but the scree plot (Figure 2) displayed inflexions that did not support the retention of all factors. Given that Kaiser’s average communality of 0.68 exceeded the threshold of 0.6 (set for samples sizes > 250), five factors were retained [25].

Table 4 and Table 5 show the factor loadings for both the pattern and structure matrices after the rotation. Factor 1 (34.10%), under the unified label of cardiometabolic and vascular spectrum disorders, was determined by the associations between diabetes, coronary heart disease, hypertension, peripheral vascular disease, dyspepsia, and chronic kidney disease. Factor 2 (9.19%) was labelled as genitourinary tract disorders and represented prostate disorders, cancer, and diverticular disorders. Factor 3 (8.20%), termed respiratory and vision spectrum disorders, comprised bronchiectasis, COPD, blindness, and peripheral vascular disease. Factor 4 (5.70%), ocular spectrum disorders, included glaucoma, blindness, and cancer. Finally, Factor 5 (4.54%), neurovascular and gastro-renal syndrome, included stroke and transient ischaemic attack, dementia, chronic kidney disease, and dyspepsia. In total, 43% of the sample could be assigned to at least one of these MM patterns, with a prevalence of 40.1% for the metabolic cardiovascular pattern, 18.7% for ocular spectrum diseases, 9.1% for neurovascular and gastro-renal syndrome, 3.3% for neoplasms with gastrointestinal pathways, and 1.4% for the respiratory and vision pattern.

## 4. Discussion

Advanced statistics provide us with a means to manipulate the complexity associated with non-indexed MM [29] by reducing it into meaningful formations, otherwise called associative MM patterns [18]. The present analysis revealed eight patterns: five for males (metabolic–cardiovascular, genitourinary tract disorders, respiratory and vision spectrum disorders, ocular spectrum disorders, and neurovascular–gastro-renal syndrome) and three for females (cardiometabolic and neurovascular spectrum disorders, respiratory conditions, and sensory impairments). The revelation of clinically stable MM patterns, where SNAP-MB could be regarded as key etiological determinants of multiple MM patterns, was the central narrative of the present study and a first in this field of inquiry. The main findings are summarised below and considered in the context of the literature.

### 4.1. Multimorbidity Patterns

Only the pattern for cardiometabolic–vascular was common to males and females, though with noticeable differences in their manifestations. The remaining identified patterns did not match. Consequently, only the cardiometabolic–(neuro)vascular pattern will be presented comparatively for both sexes. The remaining MM patterns are presented separately for each sex group.

#### Cardiometabolic–(Neuro)Vascular Disorders

This is the only pattern common to males and females. It shared the highest number of morbidities and had the highest prevalence for both sexes: 40.1% in males, which was twice that in females (21.4%).

The clinical value of the specific pattern is well acknowledged in the medical literature, encompassing morbidities that usually co-exist within a complex network of pathological pathways involving chronic inflammation and insulin resistance: diabetes, coronary heart disease, atrial fibrillation, hypertension, chronic kidney disease, peripheral vascular disease, stroke and transient ischaemic attack, dyspepsia, diverticular disease, and dementia [30,31,32].

Coronary heart disease (CHD) is closely related to peripheral vascular disease (PVD) and hypertension [33], and has a bidirectional relationship with chronic kidney disease (CKD); together, they are included among the highest risk factors for cardiovascular events [34]. Diabetes is associated with CHD, PVD, stroke [35], diverticular disease [36], cognitive decline [37], and dementia. In turn, dementia is associated with cardiovascular diseases and hypertension [38]. Hypertension is an important risk factor for both PVD [39] and CKD [37,40], alongside diabetes [41]. Finally, while cancer has been associated with many of the included morbidities like diverticular diseases [42], cardiovascular diseases [43], diabetes, and CKD [44], as part of the cardiometabolic–vascular MM pattern, it only featured for females in these analyses.

The progressive pathophysiology related to most included morbidities followed the expected pattern identified in the literature [18]. With regard to age parameter, the older group (67+ years) in both sexes suffered the most, -although females (75.6%) showed a higher association than males (62.2%). Conversely, the association of the specific MM pattern was more evident in males (34.8%) than females (21.7%) in the middle-aged group (46–66 years). Evidence from numerous studies clearly implicates various SNAP-HRBs as key modifiable determinants for most morbidities included within the specific MM pattern. Specifically, Hackshaw et al.’s [45] meta-analysis showed that even a minimal amount of smoking is related to developing CHD, while Lee et al. [46] found that a lack of physical activity accounts for 6% of CHD incidents. Similarly, Bhupathiraju and Tucker [47] clearly stated the preventative nature of a healthy diet, minimising the risk for CHD. In conclusion, it is also noteworthy to mention the contribution of SNAP-HRBs to the development of CHD. Despite the Zhao et al.’s [48] meta-analysis not revealing a positive association between moderate alcohol consumption and CHD, they did find that former drinkers exhibit an increased risk of developing CHD.

Similarly, SNAP-HRBs have been associated with the development of various morbidities within the cardiometabolic–(neuro)vascular pattern. Smoking behaviour was found to be linked with CKD in [49], where low-protein and low-phosphorous diets are also implicated [50]. In the case of dementia, an association with SNAP-HRBs was observed [51,52] with some studies suggesting that SNAP-HRBs are among the top risk factors [53]. Furthermore, the long-term effects of poor nutritional habits [54] and alcohol consumption [55] have been associated with PVD. As for diverticular diseases, protective associations were identified with physical activity [56] and vegetarian-based diets high in fibre [57]. Evidence also suggests a strong linkage between SNAP-HRBs and stroke, where, in conjunction with diabetes, hypertension and psychological and cardiac causes account for more than 90% of incidences. In particular, SNAP-HRBs are recognised as key contributors to the development of diabetes and hypertension [53,58,59,60].

### 4.2. Females

#### 4.2.1. Respiratory Conditions

This pattern accounts for only 0.4% of the sample, but its clinical value is noted because it highlights the commonalities between COPD and bronchiectasis. These two main morbidities involve the progressive damage of the airways [61], with clinical symptoms of heavy cough, sputum production, recurrent respiratory infections, and dyspnoea [62]. Several studies have identified their overlap and suggest a unified preventive and management approach [63].

The literature has convincingly shown SNAP-HRBs to be key modifiable risk factors for both COPD and bronchiectasis [64]. Smoking [65] and excess alcohol [66] have been found to increase the risk of developing the morbidities included in the respiratory disease pattern, while physical activity can help to prevent the development of respiratory disease and comorbidities [67]. Furthermore, Muralidharan et al. [68], examining the combined effects of smoking, excessive alcohol consumption, and physical activity, reported a synergistic impact on the development of the progression of respiratory diseases.

The pattern’s prevalence rates clearly indicate the progressive age-related deleterious effect of SNAP-HRBs on the development of this specific pattern. The older group (67+ years) accounted for 73.3%, the middle-aged group (46–66 years) comprised 20%, and the younger group accounted for 6.7%.

#### 4.2.2. Sensory Impairment

This is a term used by several researchers [69] and the WHO [70] to refer to a range of conditions that affect sensory functions, such as glaucoma, dementia, and hearing loss. It accounted for 3.2% of the study’s female population, and most prominently in the oldest age group (67+ years; 86.3%). However, a clear indication is that the onset of the accumulative impact of SNAP-MB on sensory impairment may be rooted in middle age, since 13% of those in the middle-aged group (46–66) were found to have developed this MM pattern compared with only 0.8% of the younger age group (18–45). Several studies have evidenced the strong associations between these morbidities [71,72,73,74]. Furthermore, the etiological factors that have been suggested, apart from age-related biological pathways [75,76,77], include those of chronic inflammation and vascular function [78], implicating the contribution of SNAP-HRBs to the development and progression of morbidity patterns.

Several studies have provided evidence supporting this argument. For example, smoking has been found to have a strong association with dementia [79], age-related macular degeneration (the leading cause of severe and irreversible vision loss) [78,80], primary open-angle glaucoma [81], and hearing loss [74,82]. Additionally, excessive alcohol usage has been found to increase the risk of hearing loss [83] and dementia [84]. Nevertheless, while evidence for glaucoma is unclear [85], a recent meta-analysis [86] found a borderline though significant association between excessive alcohol usage and glaucoma. Finally, the identification of health beneficial properties of physical activity to protect against the development of age-related degeneration [87] and its progression [88] further support the argument regarding the impact of SNAP-HRBs on this sensory impairment pattern.

### 4.3. Males

#### 4.3.1. Genitourinary Tract Disorders

This pattern is among those characterised as sex-specific, affecting 3.3% of the male sample population. The title given to this specific pattern is an accepted medical term used to unify morbidities and cancers that affect the organs of the urinary and reproductive system, including the prostate gland (prostate disorders) and colon (where diverticular disease occurs) [42]. Depending on the study’s sample and related literature, middle age seems to be the onset point [57,89,90]. Within this study, the middle-aged group (46–66) accounted for 19.3% of those who developed this specific MM pattern, with the older group (67+ years) accounting for the rest (80.7%).

The present findings align with the literature [57] and indicate that, apart from age, SNAP-HRBs seem to have an important role in the prevention, development, and progression of the included morbidities. The evidence, albeit based on single SNAP-HRBs, supports this argument. For the development and progression of benign prostatic hyperplasia, smoking emerges as a significant factor [91]. Additionally, a healthy diet plays a crucial role by influencing its pathophysiology [92] and impacting prostatic growth [88]. Evidence on alcohol consumption is mixed, mainly depending on the amount and pattern of drinking. Some studies identified the detrimental effects of alcohol on benign prostate hyperplasia, while others did not [93]. For prostate cancer, a dose–response association with pack-years of smoking [90,94] and amount of alcohol consumption has been revealed [48,95]. Finally, on the one hand, being a smoker increases the risk of developing diverticulitis by 46% in comparison to non-smokers, as a metanalysis showed [96]; on the other, being physically active [74] or having a diet higher in fibre and vegetables [97] appear to be protective factors for males of middle age or older.

#### 4.3.2. Respiratory and Vision Spectrum Disorders

This pattern accounted for 1.4% of the sample’s male population, indicating a unified framework that encompasses bronchiectasis, chronic obstructive pulmonary disease (COPD), blindness, and PVD. The main emphasis regards the impact of these morbidities on multiple organ systems and highlights the significance of understanding the possible underlying mechanisms and common risk factors. For example, COPD and bronchiectasis are both chronic respiratory diseases [98] with evidence to suggest that they often coexist [63]. Their main difference is that while bronchiectasis primarily affects the larger airways, COPD primarily affects the smaller airways [65]. Blindness refers to the loss of visual function that may arise from various causes [99]. Respiratory [100] and vascular diseases [101] are regarded to be among these causes, mainly due to their subtle and often overlooked interconnection [102]. The present study indicated that age is a key parameter, with 91.3% of those who have developed the specific MM patterns belonging to the oldest age group (67+ years). The remainder (8.7%) were in the middle-aged group (46–66 years), with none from the youngest group (18–45 years).

Several studies have also shown the impact of SNAP-HRBs as independent modifiable risk factors on the development and progression of the included morbidities, alongside other common risk factors like genetic predisposition and chronic inflammation [62]. Smoking, in particular, is well established as the most preventable risk factor for both respiratory [103,104] and peripheral vascular diseases [105]. It has also been found to contribute to age-related macular degeneration (AMD), the leading cause of blindness in older adults [80]. On the contrary, physical activity has been found to be a crucial protective factor for developing COPD [106] and peripheral vascular disease [107]. Finally, evidence regarding alcohol seems to depend on the amount and pattern of drinking [108]. Several studies found no harmful effect of moderate alcohol use regarding either peripheral vascular disease [109,110] or COPD [106]. High levels of intoxication, usually derived from chronic excessive alcohol consumption, are associated with numerous ocular [111] and vascular morbidities that may result in blindness or even death, respectively [108].

#### 4.3.3. Ocular Spectrum Diseases

This was the second most prevalent MM pattern among males in the sample at 18.7%. This pattern was labelled to reflect morbidities that involve visual impairment. Specifically, glaucoma is a group of progressive optic neuropathies characterised by the degeneration of retinal ganglion cells that leads to visual field loss, and has been identified as one of the leading causes of blindness [112]. Blindness refers to severe visual impairment caused by various factors, including glaucoma and cancer [99]. Age seems to play a crucial role. Those in the older age group (67+ years) accounted for 73% of those with ocular spectrum MM, while for the younger age group (18–45 years), it was only 1.2%. An indication of an onset point is regarded to be middle age (46–66), where a large proportion (25.8%) of the males in this age group were found to have developed this specific MM pattern.

Several studies examining ocular morbidities in the middle- and older-aged populations support this argument. For example, smoking has consistently been suggested as an independent modifiable risk factor for several eye diseases, with a dose–response effect [113]. Specifically, smoking has been found to increase the risk of glaucoma development and progression [85], vision loss [114], age-related macular degeneration (AMD) [115], and uveal melanoma [116]. Additionally, studies that examined the relationship between alcohol consumption and glaucoma have yielded mixed results. A recent meta-analysis identified a positive association between any use of alcohol and open-angle glaucoma (OAG) (OR = 1.02–1.36; CI 95%, *p* = 0.03; I2 = 40.5%), but with low-confidence evidence [86]. Evidence for physical activity is limited but suggests that it may be an important underestimated modifiable risk factor for developing glaucoma [117], mainly due to the neuroprotective effects of physical activity by improving ocular blood flow and reducing intraocular pressure [118]. Finally, diets low in retinol equivalents (e.g., vitamins A and B1), along with high magnesium intake, have been associated with an increased risk of developing open-angle glaucoma [119].

#### 4.3.4. Neurovascular and Gastro-Renal Syndrome

This pattern was observed in 9.1% of the sample’s male population. The factor name acknowledges the interconnectedness of the included morbidities and emphasise their shared etiological factors. Dementia and stroke are categorised as neurovascular conditions [120,121], while dyspepsia and CKD are classified as metabolic disorders [122]. There is evidence of clinical association between the two groups of patterns [123], showing similar pathological mechanisms, such as chronic inflammation, for neurovascular and metabolic disorders [124]. Age is also an important parameter for this group of neurovascular and gastro-renal morbidities. In the present analyses, the majority (82.1%) of those influenced by the specific pattern were from the older male group (67+ years), while the younger group (18–45) accounted for just 0.9%. There is also compelling evidence of an important role for SNAP-MB as modifiable risk factors for the development of morbidities, including within this MM pattern. It seems that the middle-aged group (46–66) marks the onset of SNAP-HRBs’ deleterious effects, as 17% of the middle-aged males in this study were found to have developed this specific MM pattern.

Numerous studies have shown that SNAP-HRBs influence the development of these types of morbidity. Smoking is a well-known risk factor for dementia [125], stroke [126], and chronic kidney disease [49], producing neurodegeneration [127], vascular damage [128], and impaired renal function [41], respectively. Similarly, excessive alcohol is associated with an elevated risk of dementia, affecting brain structure and function [129,130]. It also increases the risk of stroke, contributing to the generation of ischaemic events [131]. A well-acknowledged consequence of prolonged alcohol misuse is the development of chronic kidney disease [132].

Again, lack of physical activity is also associated with an increased risk of dementia, impaired cognitive functioning [133], and stroke due to its contribution to hypertension, which may increase cerebrovascular events [46]. Finally, poor nutrition and high-fat diets have been shown to contribute to cognitive decline, increasing the risk of dementia [52] and stroke [134], while high-sodium diets increase the risk of chronic kidney disease [50].

#### 4.3.5. Comparison with Other Studies

Directly comparing the present study’s patterns with those of other studies is challenging, mainly due to the high heterogeneity in study designs (e.g., study population, included morbidities, data sources) and implementations (e.g., statistical analyses). The present study is among the few, like Prados-Torres et al. [18], that have investigated MM patterns and were not limited to older adults in an effort to increase their clinical value. Most studies [23,135,136] focused on the older adult population, thus possibly overestimating morbidity correlations. Like most other studies, the present analyses focused on a finite number of morbidities [29]. However, while other studies used the ICD-10 to examine disease categories [22,135], we used the list of 40 morbidities derived from Barnett et al. [24] and defined by Read codes in the clinical coding system used in UK general practices. Furthermore, in line with most studies, the present analyses examined MM patterns in both sexes. Few have conducted single-sex studies (e.g., Jackson et al. [28] examined MM patterns in older women).

The data source is another field of heterogeneity. This study, among others [18,23], used primary care EHRs as the main data source. Others focused on the general population [135] or specific samples; e.g., Cornell et al. [29] focused on veterans; Schäfer et al. [22] focused on a statutory health insurance company; Jackson et al. [28] used a sample of an Australian longitudinal study. Finally, this study is among those [18,22,28,136] that used EFA as their primary analytical method, rather than cluster analyses [29,135].

Despite the design heterogeneity, some of the identified patterns resemble those of previous studies. Specifically, the cardiometabolic–vascular pattern (identified in both sexes) seems to be the most consistently observed and dominant pattern and, as such, has important clinical value. Despite minor variabilities, it has been identified in most studies that examined MM patterns [18,23,28,137,138]. Prados-Torres et al.’s [18] systematic review found a specific pattern in 10 out of the 14 studies examined. As identified here, the respiratory patterns matched the patterns that emerged in previous studies. Two studies, Holden et al. [137] and Jackson et al. [28], also found a respiratory pattern in their populations. Finally, Holden et al.’s [137] gastrointestinal and cancer pattern closely resembled the one reported in the present study, called genitourinary tract disorders, and was identified in the sample’s male population. Both patterns share gastrointestinal disorders and cancer. In the present study, prostate disorder is also included in the pattern, thereby influencing the pattern’s name. Neither psychological nor mechanical–musculoskeletal patterns were identified in the present study, despite being suggested by the systematic review of Prados-Torres et al. [18] as a frequent occurrence.

#### 4.3.6. Strengths and Limitations of the Study

The inclusion of a large number of participants and morbidities are study strengths. This was augmented with the usage of EHRs that allowed the extraction of high-quality data in relation to the sample. EFA provided further rigour to study and was preferred to cluster analysis, as it allows morbidities to interact with each other and permits a single morbidity to exist in different patterns [22]. EFA was also an efficient statistical method for tackling MM’s complexity. Following the recommendations of Osborne et al. [26], EFA revealed a concise picture of limited numbers of significantly stable and clinically valuable MM patterns resistant to possible confounding influences of inaccuracies that may follow doctors’ diagnoses or lifestyle recommendations. Moreover, the inclusion in the analysis of only highly prevalent morbidities (>1%) is paired with the high rate of cumulative variance explained by the extracted factors (56.69% for females and 61.75% for males). This is followed by an adequate goodness of fit regarding the sampling accuracy (KMO values of 0.80 for females and 0.68 for males). Additionally, the inclusion of factors of only those morbidities with eigenvalues above 1% and with factor score thresholds of 0.30 (as the minimum acceptable value for a clinical and statistically significant correlation between morbidities) provides further support to the abovementioned argument.

Another added value of EFA is the formation of easily interpretable factors that produce clinically useful results. It is notable that even when two morbidities formed a pattern (respiratory), this was based on Osborne et al.’s [26] suggestion that a factor with only two morbidities can be accepted when the morbidities have high factor loadings and are conceptually related (as was this case with COPD and bronchiectasis).

The study’s limitations are recognised. First, while the number of morbidities in the present study was considerable, the list may not have been exhaustive and important morbidities may have been missed. For example, obesity was not included in the list provided by Barnett et al. [24] and, thus, it was not possible to identify it within the analyses of the present study. However, obesity has been consistently associated with various patterns in previous studies, such as the musculoskeletal pattern [18,29].

Second, there are limitations of the EHRs themselves and how doctors or primary care staff recorded morbidities or lifestyle behaviour. It could be argued that, due to a lack of a rigorous unifying recording system for SNAP-HRBs, their vulnerability to over- or underrepresentation cannot be ignored. Furthermore, diagnoses of specific morbidities that usually play secondary roles may also be underreported in patients’ EHRs in comparison to primary conditions. This may come as a result of the mono-morbid healthcare system’s treatment protocols that are primarily focused on more “serious” or “urgent” patient morbidities that usually need periodic re-examinations.

Finally, researchers like Schäfer et al. [22] argued that excluding people without MM from EFA could produce an overestimation of the correlation between morbidities, biasing the correlation matrix. However, the counterarguments are also persuasive. For example, studying the specific population may provide a better understanding of the complex interplay between SNAP-HRBs and their associations with various morbidities. Eventually, this process may reveal a shared aetiology, since specific SNAP-HRBs share common underlying causes or mechanisms. By focusing on people with MM who have engaged with SNAP-HRBs, the emerging patterns may reflect these shared etiological factors, uncovering novel associations and pathways that contribute to the development of MM patterns. Furthermore, the identified MM patterns may be more relevant and generalisable to high-risk populations, e.g., young adults who engage in SNAP-MB.

## 5. Conclusions

The confirmed narrative of the present study is confirmation that SNAP-MB are key determinants of MM patterns that hold clinical and academic research value. The recognition of patterns (i.e., as cardiometabolic–vascular or respiratory patterns), although possibly anticipated from the existing literature, does not diminish the significance of current evidence regarding well-established associations between specific SNAP-HRBs (such as smoking) and particular morbidities (such as COPD). Other emerging MM patterns, such as “respiratory and vision spectrum disorders”, can challenge the current understanding of how the included morbidities may be interconnected other than purely statistically. However, even for this pattern, Houben-Wilke et al. [102] argued that there may be an under-investigated interconnection between morbidities, suggesting the need for more intense efforts on the aetiological research of MM patterns and their association with SNAP-MB.

No research was found that examined the accumulated impact of SNAP-MB on the (multi)morbidities included within the emerging patterns, apart from some for respiratory conditions. This highlights an important gap in the current knowledge. Indicatively, prominent figures in MM research, such as Loprinzi [1], have proposed MM-SNAP as a new multidisciplinary framework for future clinical practice and research.

Only one pattern was associated with the cardiometabolic–vascular cluster of morbidities, exhibiting noticeable sex differences in manifestation despite alignment with previous studies [18,22]. This observation suggests the possible existence of different determinants or, where similarities exist, differences in the magnitude of the effects [28].

Finally, the implication for the healthcare system is clear. There is a need to shift from single disease-based clinical practice guidelines to a more person-centred approach—an approach that will put the healing relationship between the healthcare provider and patients with MM at the centre, and where MM and SNAP-MB are at the heart of patient complexity inquiry.

## Figures and Tables

**Figure 1 ijerph-22-00485-f001:**
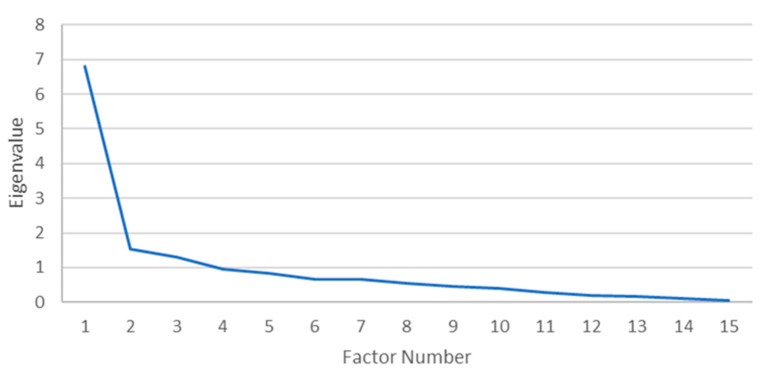
Scree plot for females.

**Figure 2 ijerph-22-00485-f002:**
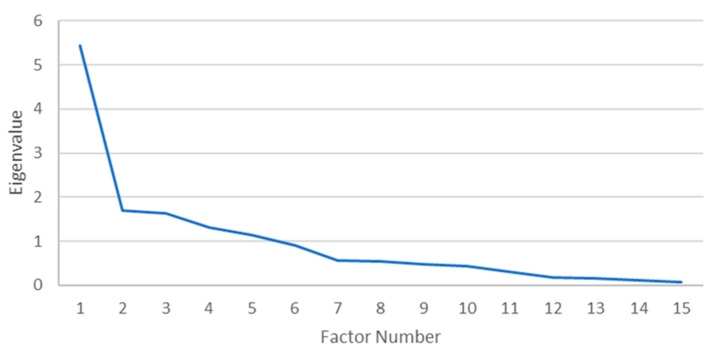
Scree plot for males.

**Table 1 ijerph-22-00485-t001:** Prevalence of morbidities by sex.

	Males		Females		Total	
Morbidities	N	%	N	%	N	%
Atrial fibrillation—AF	246	7.1	177	4.3	423	11.4
Heart failure	116	3.3	75	1.8	191	5.1
Hypertension	1628	46.8	1665	40.8	3293	87.6
Peripheral vascular disease—PVD	118	3.4	49	1.2	167	4.6
Stroke and transient ischaemic attack—Stroke TIA	238	6.8	203	5.0	441	11.8
Coronary heart disease—CHD	496	14.2	209	5.1	705	19.3
Asthma	854	24.5	1059	26.0	1913	50.5
Bronchiectasis	38	1.1	45	1.1	83	2.2
Chronic sinusitis	79	2.3	114	2.8	193	5.1
Chronic obstructive pulmonary disease—COPD	206	5.9	178	4.4	384	10.3.
Blindness	58	1.7	68	1.7	126	3.4
Glaucoma	203	5.8	204	5.0	407	10.8
Cancer	166	4.8	186	4.6	352	9.4
Prostate disorder	424	12.2	n/a	n/a	424	12.2
Chronic liver disease—CLD	139	4.0	143	3.5	282	7.5
Constipation	151	4.3	215	5.3	366	9.6
Diverticular disease	146	4.2	266	6.5	412	10.7
Dyspepsia	1687	48.5	1759	43.1	3446	91.6
Inflammatory bowel disease—IBD	442	12.7	510	12.5	952	25.2
Irritable bowel syndrome—IBS	296	8.5	703	17.2	999	25.7
Alcohol problems	144	4.1	77	1.9	221	6.0
Anorexia–bulimia	3	0.1	36	0.9	39	1.0
Anxiety	454	13.0	834	20.5	1288	33.5
Dementia	66	1.9	96	2.4	162	4.3
Depression	799	23.0	1440	35.3	2239	58.3
Schizophrenia	65	1.9	75	1.8	140	3.7
Epilepsy	92	2.6	86	2.1	178	4.7
Migraine	34	1.0	117	2.9	151	3.9
Multiple sclerosis—MS	11	0.3	34	0.8	45	1.1
Parkinson’s disease	34	1.0	22	0.5	56	1.5
Hearing Loss	973	28.0	857	21.0	1830	49
Chronic kidney disease—CKD	285	8.2	338	8.3	623	16.5
Painful condition	584	16.8	937	23.0	1521	39.8
Psoriasis and eczema	137	3.9	165	4.0	302	7.9
Rheumatoid arthritis	43	1.2	123	3.0	166	4.2
Diabetes	644	18.5	486	11.9	1130	30.4
Thyroid	210	6.0	753	18.5	963	24.5

**Table 2 ijerph-22-00485-t002:** Pattern matrix—Factor score for females’ multimorbidity patterns.

	Factor 1	Factor 2	Factor 3
	Cardiometabolic and Neurovascular Spectrum Disorders	Respiratory Conditions	Sensory Impairment
Coronary heart disease	**0.924**		
Atrial fibrillation	**0.828**		
Hypertension	**0.794**		
Peripheral vascular disease	**0.710**		
Chronic kidney disease	**0.705**		
Stroke and transient ischaemic attack	**0.702**		
Diverticular disease	**0.614**		
Diabetes	**0.546**		
Dementia	**0.440**		**0.400**
Cancer	**0.359**		
COPD		**0.820**	
Bronchiectasis		**0.787**	
Blindness			**0.790**
Glaucoma			**0.686**
Hearing loss			**0.445**

**Table 3 ijerph-22-00485-t003:** Structure matrix—Factor score for females’ multimorbidity patterns.

	Factor 1	Factor 2	Factor 3
	Cardiometabolic and Neurovascular Spectrum Disorders	Respiratory Conditions	Sensory Impairment
Hypertension	0.908		0.662
Chronic kidney disease	0.897	0.350	0.705
Atrial fibrillation	0.852	0.325	0.475
Coronary heart disease	0.824		0.392
Stroke and transient ischaemic attack	0.752		0.490
Dementia	0.689		0.661
Peripheral vascular disease	0.659		
Diverticular disease	0.635		0.380
Diabetes	0.624		0.457
Cancer	0.396		
COPD		0.829	
Bronchiectasis		0.791	
Blindness	0.433		0.786
Glaucoma	0.415		0.704
Hearing loss	0.353		0.503

**Table 4 ijerph-22-00485-t004:** Pattern matrix—Factor score for males’ multimorbidity patterns.

	Factor 1	Factor 2	Factor 3	Factor 4	Factor 5
	Cardiometabolic and Vascular	Genitourinary Tract Disorders	Respiratory and Vision Spectrum Disorders	Ocular Spectrum Disorders	Neurovascular and Gastro-Renal Syndrome
Diabetes	**0.789**				
Coronary heart disease	**0.721**				
Hypertension	**0.649**				
Peripheral vascular disease	**0.526**		**0.314**		
Dyspepsia	**0.498**				**0.456**
Chronic kidney disease	**0.488**				**0.366**
Prostate disorders		**0.860**			
Cancer		**0.654**		**0.364**	
Diverticular disease		**0.575**			
Bronchiectasis			**0.762**		
COPD			**0.623**		
Glaucoma				**0.685**	
Blindness			**0.368**	**0.524**	
Stroke and transient ischaemic attack					**0.634**
Dementia					**0.632**

**Table 5 ijerph-22-00485-t005:** Structure matrix—Factor score for males’ multimorbidity patterns.

	Factor 1	Factor 2	Factor 3	Factor 4	Factor 5
	Cardiometabolic and Vascular	Genitourinary Tract Disorders	Respiratory and Vision Spectrum Disorders	Ocular Spectrum Disorders	Neurovascular and Gastro-Renal Syndrome
Coronary heart disease	0.841	0.395			0.623
Hypertension	0.777	0.361		0.383	0.375
Diabetes	0.746				
Chronic kidney disease	0.697	0.415	0.347	0.315	0.580
Peripheral vascular disease	0.659		0.445		0.320
Dyspepsia	0.576				0.564
Prostate disorders		0.915		0.313	0.451
Cancer		0.718		0.470	0.306
Diverticular disease		0.548			
Bronchiectasis			0.727		
COPD	0.405		0.683		
Glaucoma				0.703	
Blindness	0.407		0.428	0.572	
Stroke and transient ischaemic attack	0.403			0.419	0.729
Dementia					0.623

## Data Availability

The data presented in this study are available on request from the corresponding author due to anonymised data.

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
