# Peer review of "Sex-Specific Multimorbidity–Multibehaviour Patterns in Primary Care Populations"

_ijerph, 2025, doi:10.3390/ijerph22040485_

Round 1
Reviewer 1 Report
Comments and Suggestions for Authors
The study tackles an increasing challenge for healthcare systems, the study population seems sufficient, and the statistical analysis provides interesting results.
The low percentage of COPD (table 1) in SNAP-MB, despite the high prevalence of population over 46 years old, is questionable and does not correspond to the growing numbers reported worldwide by GOLD and WHO. I suppose the diagnosis could be covered by co-existing asthma and included in the asthma group. Interestingly, COPD appears in respiratory patterns for both females and males (tables 2 and 3). Could you provide any explanation? In my opinion, this needs some discussion in the article. Was there no correlation found between cardiovascular and respiratory diseases? I suggest to include some articles from EpiChron, Chrodis etc.(f.egz. I. Ioakeim-Scoufa et al. doi:10.3390/ijerph17124242 etc.).
Overall the article is interesting and provides some ideas for further investigations.
Author Response
Comment 1: Low percentage of COPD
“The low percentage of COPD (table 1) in SNAP-MB, despite the high prevalence of population over 46 years old, is questionable and does not correspond to the growing numbers reported worldwide by GOLD and WHO. I suppose the diagnosis could be covered by co-existing asthma and included in the asthma group. Interestingly, COPD appears in respiratory patterns for both females and males (tables 2 and 3). Could you provide any explanation? In my opinion, this needs some discussion in the article. Was there no correlation found between cardiovascular and respiratory diseases? I suggest to include some articles from EpiChron, Chrodis etc.(f.egz. I. Ioakeim-Scoufa et al. doi:10.3390/ijerph17124242 etc.).”
Response:
We appreciate reviewer’s comments and acknowledge the concern regarding the relatively low percentage of COPD cases in our study despite the known high prevalence of COPD in individuals over 46 years old. However, it is important to note that our study sample includes primary care patients with multimorbidity aged 18 years and older, and not exclusively older adults, which may have contributed to a lower overall prevalence of COPD in our cohort. Additionally, COPD is a disease with a typically late onset, often developing after prolonged exposure to risk factors such as smoking, which may also be reflected in our study, since COPD may not have yet significantly affected the younger age group in the study.
Additionally, our reported prevalence (10.3%; Table 1) aligns with recent global estimates, which indicate a COPD prevalence between 9% and 12% based on the Chronic Obstructive Lung Disease (GOLD) definition (Boers et al., 2023). While global reports, such as those from WHO, estimate COPD prevalence exceeding 12% (Varmaghani et al., 2019), variations between studies can result from differences in diagnostic criteria, population demographics, and regional factors. Regarding the possibility that COPD diagnoses may be masked by co-existing asthma, we recognise this as a potential factor. However, in our dataset derived from the Electronic Health Records of the participating GP practices, COPD was classified distinctly using specific Read codes.
The presence of COPD in respiratory patterns for both males and females (Tables 2 and 3) reflects its known association with broader respiratory conditions rather than an underestimation of prevalence. This is also reflecting the onset parameter related with the COPD and the chronic exposure to SNAP-multibehaviours that may have not affected the study’s younger age group. Both patterns where COPD plays crucial role are discussed in detail in the article’s Discussion section.
Regarding the correlation between cardiovascular and respiratory diseases, prior research has shown inconsistent findings. Some studies, such as Holden et al. (2011), identified distinct clustering of respiratory conditions (e.g., COPD, asthma, and allergies) separate from cardiovascular diseases, while others (Prados-Torres et al., 2012) reported a COPD-cardiovascular association only in men aged 45–64. Similarly, Abad-Diez et al. (2014) did not include COPD in any of the three major identified cardiometabolic patterns. Our results are consistent with these findings, suggesting that while COPD and cardiovascular diseases share risk factors, they do not always cluster together statistically.
The authors feel that it is beyond the scope of this article to reflect all the above in the manuscript (which is already quite long) but hope that the reviewer is satisfied with these responses.
References
Abad-Díez, J. M., Calderón-Larrañaga, A., Poncel-Falcó, A., Poblador-Plou, B., Calderón-Meza, J. M., Sicras-Mainar, A., Clerencia-Sierra, M., & Prados-Torres, A. (2014). Age and gender differences in the prevalence and patterns of multimorbidity in the older population. BMC Geriatrics, 14, 75. https://doi.org/10.1186/1471-2318-14-75
Boers, E., Barrett, M., Su, J. G., Benjafield, A. V., Sinha, S., Kaye, L., Zar, H. J., Vuong, V., Tellez, D., Gondalia, R., Rice, M. B., Nunez, C. M., Wedzicha, J. A., & Malhotra, A. (2023). Global Burden of Chronic Obstructive Pulmonary Disease Through 2050. JAMA Network Open, 6(12), E2346598. https://doi.org/10.1001/jamanetworkopen.2023.46598
Holden, L., Scuffham, P. A., , Hilton, M. F., Muspratt, A., Ng, S. K., & Whiteford, H. A. (2011). Patterns of multimorbidity in older adults. Journal of the American Geriatrics Society, 59, S187. http://www.embase.com/search/results?subaction=viewrecord&from=export&id=L70990173%0Ahttp://dx.doi.org/10.1111/j.1532-5415.2011.03416.x
Prados-Torres, A., Poblador-Plou, B., Calderón-Larrañaga, A., Gimeno-Feliu, L. A., González-Rubio, F., Poncel-Falcó, A., Sicras-Mainar, A., & Alcalá-Nalvaiz, J. T. (2012). Multimorbidity patterns in primary care: Interactions among chronic diseases using factor analysis. PLoS ONE, 7(2). https://doi.org/10.1371/journal.pone.0032190
Varmaghani, M., Dehghani, M., Heidari, E., Sharifi, F., Moghaddam, S. S., & Farzadfar, F. (2019). Global prevalence of chronic obstructive pulmonary disease: Systematic review and meta-analysis. Eastern Mediterranean Health Journal, 25(1), 47–57. https://doi.org/10.26719/emhj.18.014
Kind Regards
Konstantinos Spyropoulos
Reviewer 2 Report
Comments and Suggestions for Authors
Dear authors,
Thank you very much for submitting this report. I consider this report very important regarding the knowledge on multimorbidity and its nature.
I would like to point out the following methodological issues:
- It looks like the authors have not assigned the exposure of the participants. If so, please download the proper STROBE checklist at https://www.strobe-statement.org/ and rewrite the manuscript accordingly. Otherwise, look for the proper checklist at www.equator-network.net and apply it to your manuscript.
- Provide a rationale for excluding the age as a variable in the exploratory analysis
Dear authors and editorial team,
I cannot comment on the quality of the language, until the authors have replied to the methodological issues I've raised.
Author Response
Comment 1: Exposure Assignment
"It looks like the authors have not assigned the exposure of the participants. If so, please download the proper STROBE checklist at https://www.strobe-statement.org/ and rewrite the manuscript accordingly. Otherwise, look for the proper checklist at www.equator-network.net and apply it to your manuscript."
Response:
We appreciate the reviewer’s observation regarding exposure assignment and have now added the STROBE checklist, with some changes to the structure/subheadings to clarify content. However, some of the STROBE contents are not applicable as this study is observational and exploratory in nature, applying exploratory factor analysis (EFA) to examine patterns of multimorbidity and multiple health risk behaviors (multibehaviours). For this reason, since EFA is used to identify latent structures among 40 morbidities and SNAP multibehaviours, the study does not follow the traditional exposure-outcome framework.
Notwithstanding, we acknowledge the importance of using appropriate reporting guidelines, and for that reason, and have made efforts to align with the STROBE checklist where applicable.
Comment 2: Rationale for Excluding Age in Exploratory Analysis
"Provide a rationale for excluding age as a variable in the exploratory analysis."
Response:
The primary goal of EFA was to uncover underlying patterns of multimorbidity and health risk behaviors based on the correlations between variables. Age does not fit the criteria for inclusion in factor identification because it is out of range of the scope of the study, which focused on the identification of sex- specific distinct multimorbidity - multibehaviours patterns. However, recognizing the importance of age in multimorbidity research, we present the prevalence of each identified factor by age group in the Discussion.
Comment 3 English language
We appreciate the reviewers' concerns about the English language; therefore, we inspected our manuscript more thoroughly. All linguistic amendments within the text are highlighted in red.
Kind Regards
Konstantinos Spyropoulos
Round 2
Reviewer 2 Report
Comments and Suggestions for Authors
Dear authors,
Why did you use an oblique rotation during the factpr analysis?
Comments on the Quality of English LanguageI have no additional comments.
Author Response
Comment 1
Why did you use an oblique rotation during the factor analysis?
Response:
We appreciate the reviewer's inquiry regarding utilizing oblique rotation to interpret factor loadings. Our detailed response to this query is provided in the following passage, as it has been extracted from the article's Statistical analysis section “Principal axis factoring with oblique rotation was chosen as a method for two reasons. Firstly, it has been acknowledged that the extracted patterns are limited, thus unable to fully explain the total variance when examining morbidities. Secondly, oblique allows the extracted factors to be associated with each other, which is appropriate here where morbidities could be associated [18], even if where a specific morbidity could be part of more than one MM pattern [22].” (p.5)
Kind Regards
Konstantinos Spyropoulos